

# Cross-sectional associations between body mass index, waist circumference, and multimorbidity: *Pró-Saúde* study

Fernanda Muniz de Macedo Stumpf[1], Alessandra Silva Dias de Oliveira[1], Eduardo Faerstein[2] and Cintia Chaves Curioni[1]

[1] Instituto de Nutrição, Universidade do Estado do Rio de Janeiro, Rio de Janeiro, Brazil
[2] Instituto de Medicina Social, Universidade do Estado do Rio de Janeiro, Rio de Janeiro, Brazil

## ABSTRACT

**Background**. This study aimed (1) To investigate the association between the Body Mass Index (BMI) and waist circumference (WC) with multimorbidity (MM), and (2) To identify patterns of MM and investigate the relationship between BMI and WC with specific combinations of MM (patterns of MM).

**Methods**. A cross-sectional study was conducted with 2,698 participants of the fourth phase of the Brazilian Longitudinal Study of Adult Health (*Pró-Saúde* Study). MM was defined by the presence of two or more morbidities. MM patterns were identified by exploratory factor analysis based on tetrachoric correlations. Logistic regression models were used to assess associations (odds ratios (OR) with the respective confidence intervals (CI)).

**Results**. Of the total number of participants, 39.5% were overweight and 30.0% were obese; 89.0% ($n = 1,468$) of women and 77.0% ($n = 952$) of men were abdominally obese. Indeed, 60.7% ($n = 1,635$) was identified with MM. For the category four or more morbidities, OR values of 5.98 (95% CI 4.84–7.13) and 7.48 (95% CI 6.14–8.18) were found for each point of increase in BMI, and 6.74 (95% CI 5.48–7.99) and 8.48 (95% CI 7.64–9.29) for each additional centimeter in the WC, for female and male, respectively. Five patterns of MM were identified: respiratory, osteoarticular, cardiometabolic, gastric, and thyroid diseases (56.4% of the total variance). Positive associations were found between BMI and patterns of cardiometabolic, osteoarticular, thyroid and gastric diseases (higher OR of 1.09 [95% CI 1.04–1.14]) and less pronounced between WC and patterns of cardiometabolic and osteoarticular (higher OR of 1.04 [95% CI 1.03–1.04]).

**Conclusions**. The results showed that an increase of both BMI and WC was associated with a higher number of morbidities and with patterns of cardiometabolic and osteoarticular diseases.

Corresponding author
Fernanda Muniz de Macedo Stumpf,
fermuniz@hotmail.com

## INTRODUCTION

Multimorbidity (MM) refers to the coexistence of multiple health conditions in a single individual (*Barnett et al., 2012*; *Academy of Medical Sciences, 2018*). MM can be measured in different ways, but assessing the number of morbidities involved, effects, and associations is still challenging (*Zheng et al., 2021*). Measuring MM using the number of reported

medical conditions is simple and useful, and it can assess the impact of different conditions on people's health (*Diane Zheng et al., 2021*). Studies indicate that diseases are grouped according to specific patterns (*Vetrano et al., 2018*; *Vetrano et al., 2019*; *Vetrano et al., 2020*). However, the studies that assess such patterns are still scarce, especially in Brazilian context, and discrepancies exist regarding the number and characteristics of the diseases considered, their severity levels, statistical analyses used, and other methodological aspects (*Prados-Torres et al., 2012*; *Prados-Torres et al., 2014*). For example, it is known that one disease can be a risk factor for others or be involved in their pathophysiological mechanisms. Furthermore, these MM patterns may be affected by socioeconomic factors and lifestyles, as well as eating habits and excess of weight (*Van Den Akker, Buntinx & Knottnerus, 1996*; *Prados-Torres et al., 2012*; *De Carvalho et al., 2018*). This excessive fat accumulation, especially abdominal fat is associated with important metabolic, hormonal, and inflammatory changes (*Flor et al., 2015*; *Hall et al., 2015*; *Catrysse & Van Loo, 2017*; *Dragano, Haddad-Tovolli & Velloso, 2017*; *Nájera Medina et al., 2019*). Therefore, excess of weight, particularly obesity, could be involved in the development of multiple non-communicable chronic diseases (NCDs) (*Samper-Ternent & Al Snih, 2012*; *Ezzati, 2017*; *Zhang et al., 2020*).

Most studies on MM focused on evaluating its definition and prevalence (*Willadsen et al., 2016*; *Xu, Mishra & Jones, 2017*; *Araujo et al., 2018*; *Nguyen et al., 2019*). Some of them sought to identify risk factors for MM. Among the risk factors considered, many of them included body mass index (BMI), and only a few considered waist circumference (WC) (*Jovic, Marinkovic & Vukovic, 2016*; *Leal Neto, Barbosa & Meneghini, 2016*; *Kivimäki et al., 2017*; *De Carvalho et al., 2018*; *Jawed et al., 2020*; *Zhang et al., 2020*; *Flores et al., 2021*). *Zhang et al. (2020)* observed an association between BMI, WC, and waist/hip ratio with MM in older adults. In Brazil, a recent study with a representative sample of the population found an association between increased BMI and MM but did not assess WC (*Flores et al., 2021*). *Petarli et al. (2019)* identified an increased prevalence of MM associated with increased WC. *De Carvalho et al. (2018)* found an association of BMI and WC with MM and BMI with the following patterns: cardiometabolic, oncological, mental/occupational, musculoskeletal, and respiratory diseases (*De Carvalho et al., 2018*; *Petarli et al., 2019*; *Zhang et al., 2020*). Thus, a Brazilian study that simultaneously assesses the impact of increased BMI and WC on MM, as the main exposure, considering the number of diseases involved and the patterns in which the diseases are grouped, can contribute to this field of research, especially in similar contexts.

The studies have suggested a positive association between the overweight and MM. This Brazilian study hypothesizes that excess weight and accumulation of fat in the abdominal area, represented by the BMI and WC, is associated with a greater number of morbidities grouped. Moreover, it is possible that both BMI and WC are related to specific diseases patterns. The objective of this study was to investigate the relation between BMI and WC with MM, and to identify patterns of MM and investigate the relationship between BMI and WC with specific combinations of MM (patterns of MM).

## MATERIALS & METHODS

### Study design and population

This was a cross-sectional study inserted in a prospective cohort study conducted at several university campuses in the State of Rio de Janeiro, Brazil, in 2011 and 2012, involving non-faculty civil servants, *Pro-Saúde* Study. All technical and administrative permanent staff members were invited to the study. The exclusion criteria were current non-medical leave of absence and working relocation to another institution (*Faerstein et al., 2005*).

The population of the present study included 2,698 participants of the fourth phase of the *Pró-Saúde* Study. The protocols of the 2011–2012 *Pró-Saúde* Study were approved by the Research Ethics Committee of the Institute of Social Medicine, State University of Rio de Janeiro (CAAE: 0041.0.259.000-11). All participants signed an informed consent form.

### Study variables

MM was evaluated in a self-reported way, through the following question: *Have you ever been told by any doctors you have had…? For each condition or disease, check Yes or No.* The options involved 18 chronic medical conditions: hypertension, diabetes (DM), hypercholesterolemia, acute myocardial infarction, angina, stroke, asthma, pulmonary emphysema, cholecystitis, peptic ulcer, gastritis, nephrolithiasis, arthrosis, slipped disc, repetitive strain injury (RSI), hyperthyroidism, hypothyroidism, and tuberculosis. All affirmative answers (Yes) for each condition/morbidity were considered for the creation of the MM variable. The original list of the study was maintained due to the lack of consensus as for the number of diseases to be considered for assessing the extent of MM, and according to the results of a systematic review, the prevalence was severely underestimated when studies considered a list with less than twelve conditions/diseases (*Chua et al., 2021*).

MM was evaluated from the simple count of self-reported diseases in the questionnaire applied in phase 4 of the *Pró-Saúde Study*, thus, the presence of two or more chronic health conditions was considered MM (*Hudon, Fortin & Vanasse, 2005*; *Valderas, Starfi & Sibbald, 2009*). From the morbidities count, participants were grouped as follows: with no MM (zero or one morbidity), with two morbidities, with three morbidities, and with four or more morbidities.

BMI and WC are the exposure variables of central interest. The following anthropometric measures were performed by trained personnel using portable digital scale with precision of 50 grams, stadiometer and anthropometric tape used: weight (kg), height (m), and WC (cm), measured according to Lohman's protocols, that recommend to use the midpoint between the lower margin of the last palpable rib and the top of the iliac crest (*Lohman, Roche & Martorell, 1988*). BMI (weight/height$^2$) was calculated based on the weight and height data.

In the analyses, BMI and WC were used continuously, but to characterize the study population, the following categories were used for BMI: underweight/eutrophic, BMI up to 24.9 kg/m$^2$; overweight, BMI between 25 and 29.9 kg/m$^2$; and obesity, BMI $\geq$30 kg/m$^2$ (*WHO, 2000*). The cut-off points recommended by the World Health Organization for South Americans were used for the WC, an abdominal adiposity indicator, considering abdominal obesity and increased cardiovascular risk as measured by WC $\geq$ 90 cm in

men and ≥80 cm in women (*WHO, 2000*). The investigated covariates were gender (female, male), age (in years), education (university or more, high school, elementary school), marital status (married/stable union, separated/divorced, widowed, single), race (black/brown, white, yellow/indigenous), and modified equivalent income proposed by the Organization for Economic Cooperation and Development (OECD) (*Celeste & Bastos, 2014*), which for the descriptive analyses was categorized into up to three minimum wages (MW), from three to six MW and more than six MW, and for the ordinal and binary logistic regression analyses it was used continuously, smoking (smoker, ex-smoker, non-smoker), physical activity in the last two weeks (yes or no), and consumption of fruits and vegetables (<or ≥ 4 times a week).

## Statistical analyses

A descriptive analysis was performed for the following variables: gender, age, education, marital status, race, income, BMI, abdominal obesity, smoking, physical activity, number of morbidities and, fruit and vegetable consumption, using the mean and standard deviations of continuous variables and the absolute and relative frequency of categorical variables, since all of them follow a normal distribution according to the Shapiro Wilk test.

Associations of BMI and WC and MM categories (none or one; two, three, and four or more morbidities) were estimated using ordinal logistic regression with the unadjusted and adjusted model, providing the OR and 95% CI. Univariate analyses of the covariates (gender, age, education, marital status, race, income, smoking, physical activity, number of morbidities and, fruit and vegetable consumption), were performed and those with a $p$-value < 0.2 were included in the model.

MM patterns were identified after submitting data to an exploratory factor analysis, technique used to investigate patterns for a large number of variables into fewer numbers of factors, to identify concurrent morbidities (*Ferrando & Lorenzo-Seva, 2017*). Since morbidity is a binary variable (presence or absence), factor analysis was implemented using a tetrachoric matrix, where both the variables were binary and the extraction method Robust Diagonally Weighted Least Squares (RDWLS) (*Asparouhov & Muthén, 2010*). The adequacy of the model was assessed using the Root Mean Square Error of Approximation (RMSEA), Comparative Fit Index (CFI), and Tucker-Lewis Index (TLI). According to the literature, the RMSEA values must be <0.08, with $a < 0.10$ CI, and the CFI and TLI values must be >0.90 or preferably >0.95 (*Brown, 2006*). It is suggested that composite trustworthiness values should be >0.70. However, values between 0.6 and 0.7 are acceptable, if the other quality indices of the model are adequate (*Hair et al., 2009*). Morbidities with factor loadings >0.30 in a given factor were considered to belong to that comorbidity pattern. The stability of the factors was evaluated by index $H$ (*Ferrando & Lorenzo-Seva, 2018*). However, values between 0.6 and 0.7 are acceptable, if the other quality indices of the model (CFI, TLI, and RMSEA) are adequate (*Hair et al., 2009*). Exploratory factor analysis was presented without stratification by sex, since it is a better fit to the model.

Adjusted binary logistic regressions were conducted to assess the association between BMI and WC and each MM pattern. It should be noted that each subject could belong to more than one pattern, as long as they accumulated more than two morbidities in each of

the patterns to which they belonged. A univariate model was used to assess the associations of covariates with MM and its patterns. Variables with a $p$-value <0.2 were included in the multivariate model. Regression models were stratified by sex.

Descriptive and regression analyses were performed using the SPSS V.20 (IBM) software and exploratory factor analysis was performed using the FACTOR 10.10 software (*Ferrando & Lorenzo-Seva, 2018*).

## RESULTS

In the studied population, 56.3% were female with a mean age of 51.5 years (SD = 9.0). Regarding BMI, 39.5% were overweight and 30.0% were obese. For WC, 89.0% of women and 77% of men exhibited abdominal obesity. As for the number of morbidities, the median showed a value of 2.0 (interquartile range: 2.0–3.0), and the total prevalence of MM was 60.7%. Approximately half of the participants had higher education or more (53.3%), were married (61.3%), white (50.1%), and with income above three minimum wages (73.2%) (Table 1). As for lifestyle, most were non-smokers (60.3%), did not perform any physical activity in recent weeks (57.0%), and consumed fruits and vegetables more than four times a week ($n = 1,611$, 54.9% and $n = 1,757$, 60.7%, respectively).

Table 2 shows the results of the ordinal logistic regressions for each MM category. The results showed measures of a positive association in the adjusted models, revealing the impact of covariates (age, gender, race, income, education, marital status, and fruit and vegetable consumption) on the outcome, contrasting with the unadjusted model. There was an increased odds of MM with BMI and WC for all categories of MM, with a progressive increase in the OR (dose–response gradient). Considering the category four or more morbidities, we found in the adjusted model to age, race, income, marital status, and consumption of fruits and vegetables, for each point of increase in BMI, OR values of 7 (95% CI [5.75–8.15]) for the total population, 5.98 (95% CI [4.84–7.13]) for female, and 7.48 (95% CI [6.14–8.18]) for male. For each additional centimeter in the WC, OR values of 7.48 (95% CI [6.16–8.48]) for the total population, 6.74 (95% CI [5.48–7.99]) for female, and 8.48 (95% CI [7.64–9.29]) for male. The results of the unadjusted model are shown in the Supplemental Files.

Five patterns of MM were identified, using an exploratory factor analysis, as described in the materials and methods section: respiratory diseases (asthma and emphysema), osteoarticular diseases (RSI, arthrosis, and slipped disc), thyroid diseases (hypo- and hyperthyroidism), cardiometabolic diseases (hypertension, DM, hypercholesterolemia, myocardial infarction, angina, stroke), and gastric diseases (ulcer and gastritis), which explained 56.4% of the total variance. This factorial solution showed adequate fit indices (RMSEA = 0.019; CFI = 0.967; TLI = 0.930). The composite reliability of the factors was also acceptable (>0.70) for almost all factors, except for the pattern of osteoarticular diseases (CR = 0.565). However, this pattern was maintained in this study, since the adjustment of the model, evaluated by CFI, TLI, and RMSEA was adequate, and the morbidities of this factor also presented adequate factor loadings. Table 3 shows the factor loadings of the morbidities, as well as the composite reliability and the H index.

**Table 1 Characteristics of the participants (2,698) of the *Pró-Saúde* Study, Rio de Janeiro—Brazil (2012–2013).**

| Variables | | Total | |
|---|---|---|---|
| | | *n* | % |
| Gender (*n* = 2,698) | Male | 1,178 | 43.7 |
| | Female | 1,520 | 56.3 |
| Age (years) (*n* = 2,698) | <40 years | 272 | 10.1 |
| | 40 a 49 years | 901 | 33.4 |
| | 50 a 59 years | 1,056 | 39.1 |
| | 60 a 69 years | 401 | 14.8 |
| | 70 a 79 years | 68 | 2.5 |
| Education (2,680) | University or more | 1,429 | 53.3 |
| | High School | 940 | 35.1 |
| | Elementary School | 311 | 11.6 |
| Marital status (*n* = 2,683) | Married/stable union | 1,645 | 61.3 |
| | Separated/divorced | 465 | 17.3 |
| | Widowed | 151 | 5.6 |
| | Single | 422 | 15.7 |
| Race (*n* = 2,674) | Black/ Brown | 1,302 | 48.7 |
| | White | 1,340 | 50.1 |
| | Yellow | 16 | 0.6 |
| | Indigenous | 16 | 0.6 |
| Modified equivalent income by OCDE (MW) (*n* = 2,673) | Into up to 3 | 718 | 26.8 |
| | From the 3 to 6 | 1,039 | 38.9 |
| | More than 6 | 916 | 34.3 |
| BMI (Kg/m2) (*n* = 2,608) | Underweight/eutrophic | 791 | 30.3 |
| | Overweight | 1,031 | 39.5 |
| | Obesity | 786 | 30.2 |
| Abdominal obesity (*n* = 2,661) | Yes | 2,224 | 83.6 |
| | No | 437 | 16.4 |
| Smoking (*n* = 2,678) | Smoker | 341 | 12.8 |
| | ex-smoker | 721 | 26.9 |
| | Non-smoker | 1,616 | 60.3 |
| Physical activity in the last 2 weeks (*n* = 2,680) | Yes | 1,149 | 42.9 |
| | No | 1,531 | 57.1 |
| Number of morbidities (*n* = 2,698) | 0 to 1 morbidity | 1,063 | 39.4 |
| | 2 morbidities | 614 | 22.8 |
| | 3 morbidities | 463 | 17.2 |
| | More than 4 morbidities | 558 | 20.7 |

In the adjusted model, positive associations were identified between BMI and WC and patterns of cardiometabolic diseases for BMI and WC, respectively; and less pronounced between osteoarticular diseases and BMI and WC; and thyroid disease and BMI, OR of 1.09 (95% CI [1.04–1.14]) for BMI (Table 4).

**Table 1** (*continued*)

| Variables | | Total | |
|---|---|---|---|
| | | *n* | % |
| Consumption of fruits (≥ 4 times a week) (*n* = 2,684) | Yes | 1,057 | 39.4 |
| | No | 1,627 | 60.6 |
| Consumption of vegetables (≥4 times a week) (*n* = 2,685) | Yes | 1,613 | 60.1 |
| | No | 1,072 | 39.9 |

Notes.
  *Mean (SD), age 51.5 (8.96); BMI 27.92 (5.20); WC 96.33 (12.88).
  **Median of morbidities 2.0 (interquartile range: 2.0–3.0).

**Table 2  Odds ratio adjusted and 95% confidence interval (CI) of the association between BMI and WC and the multimorbidity categories.**

| Number of morbidities | BMI | | | WC | | |
|---|---|---|---|---|---|---|
| | OR adjusted (IC95%) | | | OR adjusted (IC95%) | | |
| | Total | Female | Male | Total | Female | Male |
| 0–1 | – | – | – | – | – | – |
| 2 | 5,05 | 3.99 | 5.43 | 5.54 | 4.75 | 6.49 |
| | (3.82–6,29) | (2.87–5.12) | (4.98–6.79) | (4.25–6.84) | (3.52–5.98) | (5.08–7.89) |
| 3 | 6.07 | 5.00 | 6.56 | 6.55 | 5.76 | 7.57 |
| | (4.83–7.31) | (3.87–6.13) | (5.24–7.88) | (5.25–7.85) | (4.52–7.00) | (6.14–8.99) |
| More than 4 | 7 | 5.98 | 7.48 | 7.48 | 6.74 | 8.48 |
| | (5.75–8.25) | (4.84–7.13) | (6.14–8.18) | (6.16–8.78) | (5.48–7.99) | (7.64–9.29) |

Notes.
  *p* value < 0,001 for all variables. * adjusted model to age, race, income, education, marital status and consumption of fruit and vegetables; number of morbities 0–1 = no MM, the reference category.

## DISCUSSION

Our study confirmed the hypothesis that an increase in BMI and WC is associated with MM emphasizing the impact of the number of diseases involved and the patterns of MM. We took a more sensitive look at variables such as BMI and WC, continuously evaluating them and moving away from BMI categories.

Studies have shown that the risk of MM increases with increasing BMI, being twice as high in overweight individuals and ten times higher in individuals with class III obesity compared to eutrophic individuals (*Kivimäki et al., 2017*).

In Brazil, few studies have analyzed the association of BMI and WC with MM patterns. MM was evaluated in different ways, considering categories (number of diseases involved), and respecting disease patterns. It might be a differential, since other studies separately assessed the impact of body weight, through BMI, and few evaluated the accumulation of abdominal fat, by waist circumference, which, as already described, plays a fundamental role in the development of chronic diseases.

Results of previous studies corroborate the association found between BMI, WC, and MM, although the vast majority only assess BMI and not indicators of abdominal adiposity, such as WC (*Agborsangaya et al., 2013*; *Jovic, Marinkovic & Vukovic, 2016*; *Christofoletti, Streb & Del Duca, 2018*; *Zhang et al., 2020*). The study by *Petarli et al. (2019)* investigated

**Table 3  Factor loads of morbidities.**

| Variable | F1 | F2 | F3 | F4 | F5 |
|---|---|---|---|---|---|
| Asthma | **0.536** | 0.198 | 0.141 | −0.071 | −0.039 |
| Emphysema | **1.048** | −0.019 | 0.000 | 0.018 | 0.021 |
| RSI | 0.016 | **0.562** | −0.003 | −0.025 | 0.006 |
| Arthrosis | −0.011 | **0.542** | −0.071 | 0.148 | 0.026 |
| Slipped disc | −0.001 | **0.505** | 0.020 | 0.041 | 0.029 |
| Hyperthyroidism | −0.072 | −0.043 | **0.655** | 0.163 | −0.172 |
| Hipothyroidism | 0.068 | 0.008 | **0.839** | −0.050 | 0.069 |
| Hypertension | 0.128 | 0.059 | −0.119 | **0.660** | −0.098 |
| DM | 0.211 | −0.039 | −0.014 | **0.616** | −0.079 |
| Hypercholesterolemia | 0.179 | 0.118 | 0.132 | **0.346** | 0.047 |
| Acute myocardial infarction | −0.189 | 0.026 | 0.177 | **0.715** | 0.112 |
| Angina | −0.022 | 0.042 | 0.012 | **0.711** | 0.175 |
| Stroke | −0.017 | 0.088 | −0.030 | **0.526** | −0.168 |
| Peptic ulcer | −0.008 | −0.035 | −0.018 | 0.071 | **0.914** |
| Gastritis | 0.073 | 0.155 | 0.028 | −0.095 | **0.661** |
| Nephrolithiasis | 0.014 | 0.164 | −0.161 | 0.104 | 0.006 |
| Cholecystitis | 0.046 | 0.200 | 0.021 | 0.151 | 0.078 |
| Tuberculosis | 0.176 | −0.257 | −0.277 | 0.020 | 0.079 |
| **Composite reliability** | 0.794 | 0.565 | 0.720 | 0.772 | 0.773 |
| **H index** | 1.100 | 0.641 | 0.783 | 0.840 | 0.867 |

Notes.
Bold values represent the diseases that presented high factor loadings and were included in within each MM pattern.

**Table 4  Odds ratio and 95% confidence interval (CI) of the association of multimorbidity patterns with BMI and WC.**

| Patterns of MM[a] | BMI | | WC | |
|---|---|---|---|---|
| | Unadjusted OR (IC 95%) | Adjusted OR (IC 95%) | Unadjusted OR (IC 95%) | Adjusted OR (IC 95%) |
| Pattern 1—Respiratory diseases[b] | 1.04 (0.98–1.09) | 1.04 (0.98–1.10) | 1.01 (0.99–1.04) | 1.01 (0.99–1.04) |
| Pattern 2—Osteoarticular diseases[c] | 1.06 (1.05–1.09) | 1.03 (1.02–1.04) | 1.02 (1.01–1.02) | 1.02 (1.01–1.03) |
| Pattern 3—Thyroid diseases[d] | 0.96 (0.88–1.04) | 1.09 (1.04–1.14) | 0.98 (0.95–1.01) | 0.98 (0.95–1.01) |
| Pattern 4—Cardiometabolic diseases[e] | 1.10 (1.08–1.12) | 1.07 (1.06–1.09) | 1.04 (1.03–1.05) | 1.04 (1.03–1.04) |
| Pattern 5—Gastric diseases[f] | 1.00 (0.96–1.03) | 1.03 (1.01–1.05) | 1.00 (0.98–1.01) | 0.99 (0.98–1.01) |

Notes.
[a]Total of participants who had MM in one of the patterns 1,264 (43%).
[b]Pattern 1 ($n = 36$) adjusted model to consumption of fruits and vegetables and physical activity.
[c]Pattern 2 ($n = 384$) adjusted model to gender, age, income and race.
[d]Pattern 3 ($n = 25$) adjusted model to gender, age, consumption of fruits and vegetables and physical activity.
[e]Pattern 4 ($n = 690$) adjusted model to income, marital status, race and education.
[f]Pattern 5 ($n = 129$) adjusted model to gender, age, consumption of fruits and physical activity.

the association of WC in MM in adults (*Petarli et al., 2019*), and *Zhang et al. (2020)* assessed the WC among older adults (*Zhang et al., 2020*) and found an association between increased abdominal fat and MM. WC is considered relevant due to the role that abdominal adiposity plays in the development of NCDs (*Melo, 2011*; *Nájera Medina et al., 2019*).

The *Pró-Saúde Study* participants showed a very high WC (96.33 cm) and considering the cut-off points for cardiometabolic risk of 80 cm for female and 90 for male, it shows a greater predisposition to a higher risk of diseases, both in male and in female (*Associação Brasileira para o Estudo da Obesidade (ABESO), 2016*). Excess weight, especially when associated with increased abdominal adiposity, increases the risk of developing hypertension between 65% and 75% (*Hall et al., 2015*; *Seravalle & Grassi, 2017*). It is estimated that 85% of adult type 2 diabetics are also obese (*Ezzati, 2017*; *Chait & Hartigh, 2020*).

Studies suggested that female sex, aging, and the presence of obesity are some of the factors associated with MM (*Leal Neto, Barbosa & Meneghini, 2016*; *Kivimäki et al., 2017*; *De Carvalho et al., 2018*; *Jantsch, Alves & Faerstein, 2018*; *Nguyen et al., 2019*; *Song et al., 2019*). The present analyses, stratified by sex, showed associations of BMI and WC in both genders, however with greater association in male, which contrasts with the findings of previous studies. This result can be justified by the difference in body composition between male and female. Although female have proportionally more fat mass than male, the latter tend to have greater accumulation of abdominal fat (*Song et al., 2014*; *Bredella, 2017*; *Schorr et al., 2018*).

A systematic review found more than 97 patterns with two or more diseases; however, three groups prevailed in all analyzed studies, those of cardiometabolic, musculoskeletal, and mental health diseases (*Prados-Torres et al., 2014*). In addition, a Brazilian study conducted with the objective of defining the grouping patterns of diseases and relating them to socioeconomic and lifestyle factors found four groups of diseases (cardiometabolic/cancer, mental/occupational, musculoskeletal, and respiratory) and an association with high BMI values was also identified (*De Carvalho et al., 2018*).

It is noteworthy that the patterns found grouped two or more diseases, with a well-related physiological mechanism. The cardiometabolic pattern, in addition to grouping the largest number of diseases, was also associated with BMI and WC, corroborating the literature that shows the impact of excess weight on the pattern of diseases (*Flor et al., 2015*; *Murray et al., 2015*; *Tang, Liebeskind & Towfighi, 2017*; *Nájera Medina et al., 2019*). The pattern of osteoarticular diseases was also associated with adiposity. A population-based cohort study with 1,764,061 participants found a positive association of being overweight and the development of knee, hip, and hand osteoarthritis. In obese class II individuals, the risk of developing knee osteoarthritis was 4.7 times higher compared to eutrophic individuals (*Reyes et al., 2016*).

In the present study, we identified five patterns of MM: respiratory, osteoarticular, thyroid, cardiometabolic, and gastric diseases. It was similar to other studies, except for mental health diseases, which was not included in the list of diseases in the baseline study questionnaire. Furthermore, we observed positive associations between BMI and patterns of cardiometabolic, osteoarticular, thyroid, and gastric diseases. And less pronounced between WC and patterns of cardiometabolic and osteoarticular diseases.

In view of the above, high BMI and WC values impact the number of associated morbidities in the same individual. Therefore, in practice, monitoring the variables can contribute to the prevention of MM. In addition, the knowledge of the main disease patterns related to this increase can help public health policies and actions. The development

of longitudinal studies on the subject is encouraged, aiming to contribute even more significantly to this field of research.

## LIMITATIONS

A limitation of this study was the use of the MM variable with closed questions, yes or no, and not with medical data or medical records, leaving the perception of their health and/or diseases to the participants, which often may not mirror reality, due to the omission of certain conditions of morbidity. Although MM has been self-reported, it was worked with a pre-established list of morbidities, which does not allow considering other conditions related to mental health, very prevalent in MM studies (*Prados-Torres et al., 2014*; *Violan et al., 2014*; *Vetrano et al., 2020*) and other important chronic morbidities.

It is worth mentioning that the sample is not a true representation of the Brazilian population. Most participants were white, with incomes higher than the average for Brazilians, and had a higher proportion of obesity and higher values for WC. In addition, the cross-sectional design does not allow for affirming the causality of the associations.

## CONCLUSION

Both BMI and WC showed a positive association with MM in all MM categories and were more expressive in men. The increase in BMI or WC was positively associated with the accumulation of two or more morbidities. Of the five MM patterns found, the ones that showed an association with BMI and WC were those of cardiometabolic and osteoarticular diseases.in addition to these, those of gastric and thyroid diseases were associated with BMI. It would be required additional longitudinal research that assesses the association of adiposity indicators and MM patterns in different contexts and populations, especially when considering the diversity of definitions and methods of MM evaluation and its patterns.

Therefore, this study highlights how the increase in BMI and WC impacts MM, and MM patterns, opening the way for a more careful look at BMI and WC, which are simple tools that can be used in the prevention and /or worsening of MM.

### Funding

This study was financed by the Coordenação de Aperfeiçoamento de Pessoal de Nível Superior –Brasil (CAPES) –Finance Code 001 and had the support of the Fundação de Amparo à Pesquisa do Estado do Rio de Janeiro (FAPERJ) (process numbers E-26/210.064/2021). The funders had no role in study design, data collection and analysis, decision to publish, or preparation of the manuscript.

### Grant Disclosures

The following grant information was disclosed by the authors:

Coordenação de Aperfeiçoamento de Pessoal de Nível Superior –Brasil (CAPES) –Finance Code 001.
Fundação de Amparo à Pesquisa do Estado do Rio de Janeiro (FAPERJ): E-26/210.064/2021.

## Competing Interests

The authors declare there are no competing interests.

## Author Contributions

- Fernanda Muniz de Macedo Stumpf conceived and designed the study, performed the study, analyzed the data, prepared figures and/or tables, authored or reviewed drafts of the article, and approved the final draft.
- Alessandra Silva Dias de Oliveira conceived and designed the study, performed the study, analyzed the data, authored or reviewed drafts of the article, and approved the final draft.
- Eduardo Faerstein conceived and designed the study, authored or reviewed drafts of the article, and approved the final draft.
- Cintia Chaves Curioni conceived and designed the study, performed the study, analyzed the data, authored or reviewed drafts of the article, and approved the final draft.

## Human Ethics

The following information was supplied relating to ethical approvals (*i.e.*, approving body and any reference numbers):

This study was approved by the Research Ethics Committee of the Institute of Social Medicine, State University of Rio de Janeiro (CAE 0041.0.259.000-11).

## Data Availability

The raw data is available in the Supplemental Files.

## Supplemental Information

Supplemental information for this article can be found online at http://dx.doi.org/10.7717/peerj.14744#supplemental-information.

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
