# Peer review of "Cross-sectional associations between body mass index, waist circumference, and multimorbidity: *Pró-Saúde* study"

_PeerJ, doi:10.7717/peerj.14744_

## Round 0.1 · original submission · Major Revisions

This article addresses the association between BMI/WC and multimorbidity in a Brazilian context. Overall, the study design is appropriate; however, there are some issues that need to be clarified before we can judge the validity of the study. Below are my comments to move forward with a final decision; please make sure to attend to the additional comments provided by the reviewers.

Abstract
Line 41: body mass index should not be capitalized. The abbreviation was given at lines 42 then no need to repeat the abbreviation in the method section in lines 46-47. This has also to be applied in the main text. BMI and WC abbreviation were not explained in the text the first time they appeared on line 100.
Lines 56-60. I will discuss this later on in the results section. There was only positive association between MI and cardiometabolic and articular based on the confidence intervals (the rest the CI crosses 1. I would also carefully report that results as 1.04 is almost like 1.
Introduction:
Lines 73-76 is a very short paragraph and is talking about older vs middle aged. This is not relevant to the study. Your sample was not intended to capture middle aged individuals.

Lines 89-96 are out of nowhere and do not connect smoothly between the preceding and the following paragraph.

I agree with the reviewer that this study is not novel. I would avoid recurrent use of the word pioneering in the first two paragraphs of the discussion. You have already listed 2-3 references that measures the association with WC. This study is not innovative but replicated other studies in the Brazilian context. Is there anything that you can highlight about what this study adds to the literature? Is the Brazilian context different? Is there a different population than the literature? From the discussion, it seems the study is similar to other studies. It is ok; however don’t give the study more innovation than what it is.

Methods:
I would like to recommend the authors for excellent explanation about their variable operationalization. It helps the reader understand the results.

Lines 151-155: did the participants come to the research center and go measure using a scale. Was height and weight and WC reported or measured. Did all the universities use the same type of balance? What type of balance was used?

Statistical analysis. You mention in lines 180-181 that only variables with positive association in univariate were included in the model. Only gender age and education and marital status. However, in the tables later, you mention it was adjusted for all the variables. Please explain. The same should be explained for the MM patterns and BMI.

Results:
Was the data on MM normally distributed? It is best to describe the median rather than mean for the number of MM unless the sample was normally distributed. Please indicate that.

When it comes to giving the results for female and male. It is best to break them into two sentences. For example, lines 226-230. Either group BMI for female and male in one sentence and leave WC for another sentence. The way it is written now it lengthy and not easy to read
Table1: It is not clear why you have included the results for with MM and without MM; is this the univariate analysis. It is not needed especially that you did not provide p values. You can put that in an appendix
Is the sample representative of the Brazilian population? How come almost all of them had high WC and abdominal obesity. Most are nonsmokers!! Is the prevalence in Brazil of obesity 30%? Please mention this in the limitations section especially if the sample is not representative.

For the associations, please summarize the tables and no need to mention all the values again in the sentences. Like lines 226-230 or 224-247. Please make sure that you mention only those with narrow confidence interval not crossing 1 as showing positive association. Please indicate again why it was adjusted for all the variables. I was disappointed with the segregation of the MM patterns. It is logical that the disease was classified in their respective categories. I would have liked to see the patterns of disease in a single person. For example hypothyroid and HTN or osteoarthritis and HTN. This would have been novel and pioneering. One of the reviewers have alluded to the importance of patterns of disease in the same person.
It is intuitive to have association with arthritic condition. The patients may not be able to move and exercise. Obesity is risk for cardiovascular. Ofcourse this is cross-sectional and we can not infer causality.
Discussion
Please remove the sections about being pioneering. You can just state the main results and show the similarity to the literature. Please expand the discussion to include what the implications of the results in terms of clinical applications or future research agenda.

Again you have a very short paragraph on one sentence ( 285-287).

Limitations
There should be a major limitation section; highlighting the Brazilian context, cross sectional type of study (no causation); we don’t know the duration of the chronic disease

Reviewer 1 ·

Basic reporting

1. In the Discussion section, “Studies suggested that female sex, age, and the presence of obesity are some of the factors associated with MM …” did the authors mean “aging” or “greater age”?

Experimental design

1. In the Materials & Methods section, “This was a cross-sectional study inserted in a prospective cohort study conducted at several university campuses in the State of Rio de Janeiro, Brazil …” During what time was this study conducted? This needs to be clarified.
2. In the Statistical Analyses section, “From the identified patterns, binary variables were created for each pattern (belonging or not to each pattern).” What were the patterns and how were they identified? Please clarify.
3. In the Results section, “Five patterns of MM were identified …” are these patterns the same as those mentioned in the previous section? If so, they should be explained when first mentioned.

Validity of the findings

1. Did the authors implement any adjustments for confounding? For example, age and family history are two potential confounders that could result in both high BMI and WC, as well as morbidity.

·

Basic reporting

Abstract
Line 46-47: 'Body mass index (BMI) and waist circumference (WC) and MM was defined by the presence of two or more morbidities'. Consider revising to include definitions for BMI and WC or focus on defining MM only.

Introduction
The study focuses on Brazil. Please declare this early in the introduction, yet pointing out the international significance of the issues raised.
You alluded to MM being common in the middle-aged. Unfortunately, you seem not to have developed this point fully. If middle-age is your focus, then make it explicit.
Line 86 & 87 …patterns may be affected by socioeconomic factors and lifestyles, such as eating habits and excessive weight gain. Excessive weight gain is not an example of lifestyle but an outcome/product of lifestyle choices. Consider revising this.
Line 114 –This study hypothesises that the higher BMI and WC values the greater the number of morbidities- rewrite to make clearer
Lines 82-84 –'… studies that assess such patterns are still scarce, and discrepancies exist regarding the number and characteristics of the diseases considered, their severity levels, statistical analyses used, and other methodological aspects (Prados-Torres et al., 2012). Consider using a more recent reference to support it
Lines 256-257- please clarify what you mean by evaluating MM qualitatively. Is this reported in the methods and findings?

Experimental design

No comment

Validity of the findings

The findings section conveys a clear picture of the data. However, the discussion requires work to ensure clear communication of the results' meaning, relevance and implications.

The first two paragraphs reiterate the novel nature of your finding (first to associate the visceral adiposity indicator with MM patterns/first in Brazil). Your study investigated 1) the relationship between BMI, WC and MM and to identify patterns of MM and 2) the relationship between BMI and WC with specific combinations of MM (patterns of MM). Consider recapping this question /problem and then summarise the key findings to demonstrate how you addressed the identified gap.

Additional comments

Thank you for the opportunity to review this informative paper. Studies that explore the patterns of the grouping of diseases in the same person are scarce, with inconsistencies in the number and characteristics of diseases considered. The current study addressed this gap by assessing the impact of increased BMI and WC on MM as the main exposure, considering the number of diseases involved and the patterns in which the diseases are grouped. Overall, I found the paper mainly well-written

---

## Round 0.2 · accepted · Accept

The reviewer is satisfied with the authors' comments and modifications.

Reviewer 1 ·

Basic reporting

No comment.

Experimental design

No comment.

Validity of the findings

No comment.